# Removal of Tetracycline Hydrochloride by Photocatalysis Using Electrospun PAN Nanofibrous Membranes Coated with g-C_3_N_4_/Ti_3_C_2_/Ag_3_PO_4_

**DOI:** 10.3390/molecules28062647

**Published:** 2023-03-14

**Authors:** Peng Wang, Xu Han, Xianhong Zheng, Zongqian Wang, Changlong Li, Zhiqi Zhao

**Affiliations:** 1School of Textile and Garment, Anhui Polytechnic University, Wuhu 241000, China; 2Anhui Province College of Anhui Province College Key Laboratory of Textile Fabrics, Wuhu 241000, China; 3Key Laboratory of Intelligent Textile and Flexible Interconnection of Zhejiang Province, Zhejiang Sci-Tech University, Hangzhou 310018, China

**Keywords:** photocatalytic, PAN nanofiber membrane, degradation, tetracycline, heterojunction

## Abstract

In order to improve the photocatalytic performance of g-C_3_N_4_, the g-C_3_N_4_/Ti_3_C_2_/Ag_3_PO_4_ S-type heterojunction catalyst was prepared by electrostatic assembly method, and then the g-C_3_N_4_/Ti_3_C_2_/Ag_3_PO_4_/PAN composite nanofiber membrane was prepared by electrospinning technology. The morphology and chemical properties of the nanofiber membrane were characterized by SEM, FTIR, and XRD, and the photocatalytic degradation of tetracycline hydrochloride (TC) in water by the nanofiber membrane was investigated. The results showed that g-C_3_N_4_/Ti_3_C_2_/Ag_3_PO_4_ could be successfully loaded on PAN and uniformly distributed on the surface of composite nanofiber membrane by electrospinning technology. Increasing the amount of loading and catalyst, lowering the pH value and TC concentration of the system were conducive to the oxidation and degradation of TC. The nano-fiber catalytic membrane had been recycled five times and found to have excellent photocatalytic stability and reusability. The study of catalytic mechanism showed that h^+^, •OH and •O_2_^−^ were produced and participated in the oxidation degradation reaction of TC, and •O_2_^−^ plays a major role in catalysis. Therefore, this work provides a new insight into the construction of high-performance and high-stability photocatalytic system by electrospinning technology.

## 1. Introduction

Since 2020, a large number of antibiotics have been used to eliminate the global pandemic COVID-19. The following antibiotic pollution has become the main factor threatening human health and the balance of aquatic ecosystem, which seriously restricts the rapid development of China’s water environment ecological compensation and livelihood project [1]. There is thereby an urgent need, but it is still a significant challenge to eliminate residual antibiotics in surface water, groundwater and even drinking water.

Photocatalytic technology has the advantages of strong oxidation capacity, simple reaction conditions, high efficiency, low cost and harmless final products, and is considered to be an effective technology for treating antibiotic wastewater and reducing its drug resistance [2]. At present, the photo-catalysts used mainly include TiO_2_ [3], Ag_3_PO_4_ [2,4] and ZnIn_2_S_4_ [5]. Although these photo-catalysts have significant degradation effects on antibiotics, there are still many shortcomings [4]: Fast recombination rate of photo-generated carriers, low utilization rate of visible light, poor stability, poor recovery and reuse performance, etc. Therefore, it is urgent to develop a solar-driven catalyst for the efficient degradation of antibiotic pollutants in water. Metal-free organic photocatalysis merges organic synthesis with photochemistry and enables many challenging organic transformations to take place under ambient conditions while using photon as the sole energy source [6]. g-C_3_N_4_ is a new metal-free polymer visible light-driven semiconductor photo-catalyst [7,8,9], which can absorb visible light with wavelength less than 470 nm and has irreplaceable advantages. However, due to the small band gap width of g-C_3_N_4_, its specific surface area is too small, and its visible light utilization rate is low. Furthermore, the electron-hole recombination is fast and it is easy to agglomerate and precipitate, resulting in low photocatalytic activity [10]. In order to solve the above problems, researchers have adopted various methods to modify or redesign them, such as element doping [11], nanostructure design [12], synergetic catalysis [13] or forming hetero-junctions with other semiconductors [14]. Among the various modification methods of g-C_3_N_4_, the construction of type II hetero-junction catalyst can improve the effective separation of electrons and holes to a certain extent, thus improving its photocatalytic performance [15], which is a relatively effective modification method. However, the reduction ability of photo-generated electrons and the oxidation ability of holes in this type II hetero-junction will be significantly reduced. In recent years, a new type of solid semiconductor S-type hetero-junction photo-catalyst, compared with the traditional type II hetero-junction, can more effectively separate the generated charge and make the photo-catalyst have better oxidation-reduction ability than the single catalyst [15]. The S-type hetero-junction usually uses noble metal conductors such as Ag, Au and Pt as the electron transmission body between the two semiconductors [16]. However, these precious metals are expensive and are prone to causing heavy metal pollution. Ti_3_C_2_ is a new 2D material with similar structure to g-C_3_N_4_. It has good metal conductivity, good hydrophilicity, and can provide rich active catalytic sites for photocatalytic reactions [2,17]. In addition, Ag_3_PO_4_ has high quantum yield and strong oxidation ability under visible light radiation, and is considered as an effective photo-catalyst for degradation of organic pollutants [4]. Importantly, when most nano-scale powder or sheet photo-catalyst semiconductors are applied to water environment purification, there are problems such as reduced catalytic performance, easy loss, poor adsorption performance, secondary pollution and difficult recovery or low recovery efficiency, which lead to higher costs.

The preparation of fiber photo-catalyst film by loading catalyst on the surface of nano-fiber film through electrostatic spinning technology can not only increase its contact surface with pollutants, improve its adsorption performance for pollutants, but also solve the problems of easy agglomeration and difficult recovery of powder catalyst [18,19]. Therefore, in this study, firstly, the g-C_3_N_4_/Ti_3_C_2_ composite catalyst was constructed by electrostatic assembly, and then the Ag_3_PO_4_ was in situ generated on the surface of Ti_3_C_2_ by ion exchange method to prepare the g-C_3_N_4_/Ti_3_C_2_/Ag_3_PO_4_ composite photo-catalyst. Finally, the composite catalyst was loaded onto the surface of nano-fibers by electrostatic spinning technology to prepare nano-fiber photocatalytic membrane, and then it was applied to the oxidation degradation of Tetracycline Hydrochloride (TC) under the condition of light radiation. The catalytic mechanism of the catalyst was also studied.

## 2. Results and Discussion

### 2.1. Analysis of FTIR

There is a classical characteristic peak of PAN nano-fibers in the FTIR spectrum of PAN nano-fibers, as presented in Figure 1. An intensive absorption band at 2930 cm^−1^ corresponded to the -CH_2_, the band observed at 2240 cm^−1^ was the stretching vibration of C≡N and the peak at 1730 cm^−1^ was due to the asymmetric vibration of the C=O groups of PAN nano-fibers. The test results are consistent with Li’s research results [20]. The peaks in the region 3000–3400 cm^−1^ were due to the N-H groups, and the peaks in the region 1200–1680 cm^−1^ and 810 cm^−1^ were due to the stretching vibrations of C-N and triazine. It is noteworthy that no characteristic absorption peaks belonging to Ti_3_C_2_ and Ag_3_PO_4_ were observed in the FTIR spectra of g-C_3_N_4_/Ti_3_C_2_/Ag_3_PO_4_, which is consistent with Ding’s research results [21].

### 2.2. Analysis of X-ray Diffractometer

Figure 2 presents the XRD patterns of PAN nano-fibers and the intensity peaks at 16.33° and 22.42° correspond to typical characteristic diffraction peaks of PAN nano fibers [22]. The intensity peaks at 27.33°, 31.85° and 45.85° of g-C_3_N_4_/Ti_3_C_2_/Ag_3_PO_4_ correspond to (002), (200), (111) diffraction of g-C_3_N_4_, respectively. Some peaks also appeared at 2θ = 6.21° and 44.04°. These were detected in the X-ray diffraction pattern of Ti_3_C_2_, which were distributed to (002) and (105) diffraction of Ti_3_C_2_. The diffraction peak at 37.87° corresponds to the (211) crystal plane of Ag_3_PO_4_ in the g-C_3_N_4_/Ti_3_C_2_/Ag_3_PO_4_ catalyst. It is worth noting that the XRD spectrum of g-C_3_N_4_/Ti_3_C_2_/Ag_3_PO_4_/PAN corresponds to the obvious diffraction peak of PAN, and the diffraction peak of g-C_3_N_4_/Ti_3_C_2_/Ag_3_PO_4_ is weak. These were indicative that during electro-spinning, g-C_3_N_4_/Ti_3_C_2_/Ag_3_PO_4_ catalyst was dispersed into PAN fiber.

### 2.3. Analysis of UV-Vis Diffuse Reflectance Spectra (DRS)

Figure 3 shows and compares the optical absorption performance of PAN, g-C_3_N_4_/Ti_3_C_2_/Ag_3_PO_4_ and g-C_3_N_4_/Ti_3_C_2_/Ag_3_PO_4_/PAN. DRS spectra show that PAN nano-fibers have optical absorption properties only for wavelengths less than 230 nm. In addition, it is not difficult to find that the g-C_3_N_4_/Ti_3_C_2_/Ag_3_PO_4_ catalyst has excellent light absorption performance in the wavelength range of 200–800 nm, which provides a basic condition for preparing the g-C_3_N_4_/Ti_3_C_2_/Ag_3_PO_4_ catalyst with excellent photocatalytic performance. It is worth noting that the DRS spectrum of g-C_3_N_4_/Ti_3_C_2_/Ag_3_PO_4_/PAN is similar to that of g-C_3_N_4_/Ti_3_C_2_/Ag_3_PO_4_ catalyst, both of which have excellent light absorption properties [23]. This means that the prepared composite nano-fibers have excellent photocatalytic properties, and PAN nanofibers have no significant effect on the optical absorption properties of g-C_3_N_4_/Ti_3_C_2_/Ag_3_PO_4_ catalyst.

### 2.4. Analysis of Zeta Potential

Malvern Zetasizer Nano ZSE analyzer (ZEN3700, MALVERNa) was used to measure the Zeta potential of g-C_3_N_4_ and Ti_3_C_2_ aqueous dispersions, and the results are shown in Figure 4. The Zeta potential of g-C_3_N_4_ and Ti_3_C_2_ dispersions is 16.8 mV and −25.8 mV respectively. The Zeta potential test shows that when Ti_3_C_2_ is added to the g-C_3_N_4_ dispersion, Ti_3_C_2_ and g-C_3_N_4_ can form the g-C_3_N_4_/Ti_3_C_2_ catalyst through electrostatic attraction, and Ag^+^ can be adsorbed on the surface of Ti_3_C_2_ through electrostatic assembly, further forming the composite catalyst. This provides theoretical support for the preparation of stable composite catalysts.

### 2.5. Morphology Analysis

The SEM studies revealed the surface morphology of PAN nano-fibers and its composite catalytic film, as shown in Figure 5. The SEM images confirm that nanofibers in the PAN before composite showed vertical, random and uniform dispersion, as shown in Figure 5(a1–a2). The transmission electron microscopy (TEM) images show that the g-C_3_N_4_/Ti_3_C_2_/Ag_3_PO_4_ catalyst was successfully synthesized by electrostatic assembly in Figure 5c. After the composite of g-C_3_N_4_/Ti_3_C_2_/Ag_3_PO_4_ catalyst and nano-fibers, it can be observed that there are many granular substances uniformly distributed on the fiber surface (Figure 5(b1)), which indicates that the catalyst has been successfully fixed to the fiber surface [24]. Further enlarge the multiple of the SEM diagram. From the SEM diagram, it can be seen that the diameter of the fiber increases after the catalyst is compounded, which may be caused by the influence of the powder catalyst on the viscosity and other properties of the PAN spinning solution during the electro-spinning process, as shown in Figure 5(b2).

### 2.6. Photocatalytic Performance of Nanofiber Photocatalysis Film

Nano-fiber photocatalytic film was applied to the photocatalytic oxidation degradation of TC under different conditions, and the effect of reaction conditions on its photocatalytic performance was investigated, as shown in Figure 6. It can be seen in Figure 6a that when PAN nano-fibers exist, the R% of TC at 120 min is only 17.90%, which is mainly due to the adsorption of PAN fibers on TC molecules. In the presence of nano-fiber photo-catalysis film, the R% of TC increased significantly with time, and the R% value increased with the increase of g-C_3_N_4_/Ti_3_C_2_/Ag_3_PO_4_ loading. This indicates that the increase of g-C_3_N_4_/Ti_3_C_2_/Ag_3_PO_4_ loading on the surface of nano-fibers can promote the oxidation degradation of TC. This is mainly because the increase of catalyst loading is conducive to the decomposition of water molecules or oxygen molecules to produce more highly oxidizing •OH and •O_2_^−^ under the condition of light radiation, which increases the number of active free radicals participating in the oxidation degradation reaction of TC molecules in aqueous solution, thus increasing the R% value of TC. It is worth noting that when the loading amount of nano-fibers is more than 300 mg·g^−1^, the R% value of TC decreases in the same time. This means that excessive catalyst loading is not conducive to the oxidative degradation reaction. On the one hand, it is easy to agglomerate when the content of catalyst in the spinning solution is too high, reducing the specific surface area of the catalyst, and thus reducing its catalytic performance. On the other hand, in the process of electro-spinning, a large number of g-C_3_N_4_/Ti_3_C_2_/Ag_3_PO_4_ catalysts were coated inside the nano-fibers, which could not be effectively contacted during the oxidation degradation of TC.

Figure 6b shows the effect of pH value on the photocatalytic degradation performance of nano-fiber photo-catalysis film. It can be seen from the figure that in the range of pH = 3–9, the R% value of TC increases significantly with time, and the R% value at 120 min is higher than 70%. This means that the nano-fiber photocatalytic film has wide pH adaptability. It is not difficult to find that the R% value of TC gradually decreases with the increase of pH value at the same time. This indicates that raising the pH value is not conducive to the oxidative degradation reaction. This is mainly because under acidic conditions, TC mainly exists in the form of cationic TCH_3_^+^ [25], and the surface of nano-fibers has negative ionic groups. The electrostatic attraction enables the nano-fibers to rapidly absorb TC, thus accelerating the reaction. When the pH value increases, TC exists in the form of zwitterion (TCH_2_^0^ or TCH_1_^−^) [25], which leads to the reduction of electrostatic attraction and adsorption between nanofibers and TC molecules.

Under the condition of light radiation, the nano-fiber photocatalytic film was applied to the oxidation degradation reaction of different initial concentrations of TC to study the relationship between the concentration of pollutants and the catalytic performance of the system (Figure 6c). It can be seen from the figure that with the increase of TC concentration, its R% value gradually decreases. This means that the increase of TC concentration in aqueous solution in the presence of nano-fiber photocatalytic film is not conducive to its oxidative degradation reaction. On the one hand, the increase of the initial concentration of TC makes many TC molecules gather to form a multi-molecule complex, resulting in the increase of steric hindrance, which increases the resistance for the photocatalytic oxidation degradation of TC molecules. On the other hand, the concentration of active free radicals (•OH and •O_2_^−^) produced by the decomposition of water molecules or dissolved oxygen molecules by g-C_3_N_4_/Ti_3_C_2_/Ag_3_PO_4_ catalyst remains unchanged, and the number of active digits of the reactants adsorbed on the surface of the catalyst is limited. The increase of TC concentration leads to the reduction of the rate of its oxidative degradation reaction with active free radicals. In addition, Figure 6d shows the effect of the amount of nano-fiber photocatalytic film on the R% value. The results showed that the R% value of TC gradually increased with the increase of the amount of fiber membrane. It is noteworthy that when the fiber dosage is greater than 0.2 g, the R% value of TC will not increase significantly. This is because in the oxidation degradation reaction of TC, the light radiation intensity does not increase, that is, the number of photons exposed to the nano-fiber photo-catalysis film is certain, and the concentration of active free radicals in the solution cannot be increased by continuing to increase the amount of fiber.

To study the decomposition and destruction of aromatic ring structure and unsaturated bond in the process of catalytic oxidation and degradation of TC by nano-fiber photo-catalysis film, Lambda 950 UV-Vis-NIR spectrophotometer was used to analyze the oxidation and degradation of TC in the wavelength range of 220–500 nm, as shown in Figure 7. During the oxidation degradation reaction of TC, the characteristic absorption peak of carbonyl at λ = 357 nm and the structure of TC aromatic ring at λ = 276 nm decreased with the prolongation of reaction time, indicating that most of TC was degraded during the reaction (Figure 7). With the short reaction time, the characteristic absorption peak at λ = 276 nm shifted to 272 nm and 259 nm. These results mean that the conjugated structure of TC is destroyed and decomposed into small molecular fragments during the oxidative degradation reaction.

It is of great significance for practical application to study the reusability of nano-fiber photocatalytic film as heterogeneous photocatalytic reaction catalyst. Under the same conditions, five oxidation degradation experiments of TC were carried out on nano-fiber photocatalytic film. As shown in Figure 8, the R% value of TC did not decrease significantly with the increase of the number of times of reuse of the nano-fiber photo-catalysis film. At the fifth time of reuse, R% value at 120 min was still as high as 78.43%, which showed that the catalytic oxidation activity of the nano-fiber photo-catalysis film on TC was still high after five repetitions, and there was almost no deactivation. This means that the nano-fiber photocatalytic film can be used as a long-term stable heterogeneous catalyst for the oxidative degradation of TC in water.

### 2.7. Photocatalytic Mechanism

The adsorption of TC onto the photocatalysts had an important effect on the photodegradation of TC. So, the specific surface areas (BET) of blank PAN, g-C_3_N_4_/Ti_3_C_2_/Ag_3_PO_4_ and g-C_3_N_4_/Ti_3_C_2_/Ag_3_PO_4_/PAN have been determined by N_2_ adsorption using Micromeritics ASAP 2460 equipment, which are 10.0041 m^2^·g^−1^, 81.3913 m^2^·g^−1^ and 20.0023 m^2^·g^−1^ respectively. As displayed in Figure 9a, when the blank PAN exists and without catalyst, the concentration of TC hardly decreased with the increasing of time. The TC decreased by 4% after 120 min. This suggested that the adsorption and photo-degradability performance of blank PAN and light to TC was weak in aqueous solution. Under the same conditions, the powder catalysts g-C_3_N_4_ and g-C_3_N_4_/Ti_3_C_2_/Ag_3_PO_4_ were used for oxidative degradation of TC in Figure 9a. It shows that the R% values of g-C_3_N_4_, g-C_3_N_4_/Ti_3_C_2_/Ag_3_PO_4_ and their nano-fiber membranes to TC are 45.7%, 96.2% and 82.8% at 120 min, respectively, which means that the catalytic performance of g-C_3_N_4_ can be significantly improved through the construction of hetero-junction. Moreover, the catalytic performance of the nano-fiber photocatalytic film was not significantly affected. To further clarify the process and mechanism of the photocatalytic reaction, the effect of free radical trapping agent on the catalytic activity of the nano-fiber photocatalytic film was studied, and the main active substances for the oxidative degradation of TC were determined. As shown in Figure 9**b**, EDTA-2Na, IPA and BQ are the trapping agents of hole (h^+^), hydroxyl radical (•OH) and superoxide radical (•O_2_^−^) respectively [5,26], and their dosage is 1 mM·L^−1^. After adding EDTA-2Na, IPA and BQ into the reaction system, the R% value of TC decreased from 82.8% to 59.3%, 62.2% and 14.6%, respectively within 120 min of light radiation time. This shows that h^+^, •OH and •O_2_^−^ are produced and participated in the photocatalytic degradation reaction, and •O_2_^−^ plays a major role in the oxidative degradation of TC.

With BaSO_4_ as the reference, the optical properties of g-C_3_N_4_ and Ag_3_PO_4_ were tested by UV-Vis-NIR spectrophotometer, and according to the formula (αh*v*)^1/2^ = A(h*v* − E_g_) is calculated [27] in Figure 10a. The band gap widths of the two are 2.66 eV and 2.41 eV respectively. According to the literature calculation, the valence band (VB) and conduction band (CB) positions of [28] g-C_3_N_4_ are 1.55 eV and −1.11 eV, respectively, and the valence band and conduction band positions of Ag_3_PO_4_ are 2.675 eV and 0.265 eV, respectively. Ti_3_C_2_ as an electron transport body can build a S-type heterostructure between the three. Under the condition of light radiation, the electrons in the valence band of g-C_3_N_4_ and Ag_3_PO_4_ migrate to their conduction bands respectively, and then through the rapid transport of Ti_3_C_2_, the holes in the valence band of g-C_3_N_4_ and the electrons in the conduction band of Ag_3_PO_4_ recombine, thus inhibiting the recombination of photo-generated electrons and holes in g-C_3_N_4_ and Ag_3_PO_4_. Since the potential of the g-C_3_N_4_ conduction band is higher than the standard reduction potential of O_2_/•O_2_^−^ (−0.33 eV) [26], the photo-generated electrons retained in the g-C_3_N_4_ conduction band can reduce O_2_ to •O_2_^−^. At the same time, because the potential of Ag_3_PO_4_ valence band is lower than OH^−^/•OH (2.38 eV) [26], the photogenerated holes in its valence band can oxidize OH^−^ to •OH. In addition, some photo-generated holes in the valence band of Ag_3_PO_4_ also participate in the oxidative degradation reaction. The resulting h^+^, •OH and •O_2_^−^ subsequently participate in the photocatalytic oxidation degradation of TC. In summary, the main reaction steps of this mechanism under the condition of light radiation are as follows (1)–(5):(1)$$g-C_{3}N_{4}/Ti_{3}C_{2}/Ag_{3}PO_{4}\overset{hv}{\to }g-C_{3}N_{4}\left(e^{-},h^{+}\right)/Ti_{3}C_{2}/Ag_{3}PO_{4}\left(e^{-},h^{+}\right)$$
(2)$$g-C_{3}N_{4}\left(e^{-},h^{+}\right)/Ti_{3}C_{2}/Ag_{3}PO_{4}\left(e^{-},h^{+}\right)\;\overset{Z-scheme}{\to }\;g-C_{3}N_{4}\left(e^{-}\right)+Ag_{3}PO_{4}\left(h^{+}\right)$$
(3)$$g-C_{3}N_{4}\left(e^{-}\right)+O_{2}\to \;g-C_{3}N_{4}+•O_{2}^{-}$$
(4)$$PO_{4}\left(h^{+}\right)+OH^{-}\to Ag_{3}PO_{4}+•OH$$
(5)$$TC\overset{•\;O_{2}^{-}\;or\;OH\;or\;H^{+}}{\to }Degraded\;Products$$

## 3. Materials and Methods

### 3.1. Materials

N. N-dimethylformamide (DMF), anhydrous ethanol, Tetracycline Hydrochloride, urea, nitric acid, sodium hydroxide, silver nitrate, sodium dihydrogen phosphate, isopropanol, p-benzoquinone, etc. are all analytical reagents purchased from Aladdin Bio-Chem Technology Co., Ltd. (Shanghai, China). Polyacrylonitrile (PAN) (150,000 Da) was purchased from Sinopharm Chemical Reagent Co., Ltd. (Shanghai, China). Ti_3_C_2_ was purchased from Nanjing XFNANO Materials Tech Co., Ltd. (Nanjing, China).

### 3.2. Synthesis of Catalyst

(1) g-C_3_N_4_/Ti_3_C_2_/Ag_3_PO_4_ composite photo-catalyst: Put the specified amount of urea in a crucible, heat it up to 100 °C in a tubular furnace, and keep it for 20 min, with the heating rate of 5 °C/min; Then raise the temperature to 550 °C, the heating rate is 10 °C/min, and the temperature drops naturally to room temperature after holding for 3 h. After taking out and grinding, add 4.0 g of g-C_3_N_4_ powder into 200 mL of 0.5 mol/L nitric acid aqueous solution. After ultrasonic and vigorous stirring, centrifuge and wash it to pH = 7.0. Then, Ti_3_C_2_ with 7% mass fraction of g-C_3_N_4_ was added into the g-C_3_N_4_ dispersion under stirring conditions. Then add 30 mg of silver nitrate into the solution, stir it in dark for 2 h, and then add sodium dihydrogen phosphate. After the electrostatic assembly is completed, centrifuge the precipitate, dry it in vacuum for 12 h at 60 °C, and grind it for use.

(2) Nanofiber photo-catalyst film: 300 mg of composite photo-catalyst into the spinning solution dissolved with 1.0 g of PAN, stirring for 1 h, and ultrasonic treatment for 2 h to obtain a uniformly dispersed functional spinning solution. Subsequently, the g-C_3_N_4_/Ti_3_C_2/_Ag_3_PO_4_ nanofiber membrane was prepared by electrospinning technology. The spinning voltage was 20 kV, the receiving distance was 15 cm, and the propulsion speed was 1 mL/h.

### 3.3. Characterization

Scanning electron microscopy (SEM) images were examined with a Hitachi S4800 field emission scanning electron microscope under 20 kV accelerating voltage. The morphology was determined by transmission electron microscopy (TEM, FEI Tecnai G20). The sample structure was determined by X-ray powder diffraction (XRD, Bruker D8). Fourier transform infrared (FTIR) spectroscopy was examined by a Shimadzu IR Prestige-21 spectrometer to collect the vibration modes of functional groups within 4000–400 cm^−1^.

### 3.4. Photocatalytic Activity

Use distilled water to prepare 50 mL of tetracycline hydrochloride (TC) aqueous solution with a concentration of 20 mg L^−1^ (except for special instructions) for use. Adjust the pH value of TC solution by adding H_2_SO_4_ or NaOH. Then cut 0.2 g (except for special instructions) of nanofiber photocatalytic film and put it into the above TC aqueous solution and put it in the photo-reactor to keep it at room temperature and dark state for 2 h. After adsorption equilibrium, turn on the light source to carry out the photocatalytic oxidation degradation reaction of TC. The light source is a 300 W xenon lamp and a filter is used (λ ≥ 420 nm). After the reaction starts, at the maximum absorption wavelength of TC at certain intervals (λ = 357 nm), calculate the degradation rate (R%) of TC in the reaction process according to Formula (6):*R*% = (*C*_0_ − *C_t_*)/*C*_0_ × 100%(6)where *R*% is the degradation rate, *C*_0_ and *C_t_* are the initial concentration and time *t* concentration of TC solution.

## 4. Conclusions

The g-C_3_N_4_/Ti_3_C_2_/Ag_3_PO_4_ hetero-junction catalyst was prepared by electrostatic assembly method, and then the nano-fiber composite photo-catalyst film was prepared by electrostatic spinning technology. The effects of load, pH value, TC concentration and nano-fiber dosage on its catalytic performance were investigated, and the catalytic mechanism was researched. SEM, FTIR and XRD studies showed that g-C_3_N_4_/Ti_3_C_2_/Ag_3_PO_4_ catalyst has been successfully loaded onto the surface of nano-fibers. Increasing the amount of loading and catalyst, decreasing the pH value and TC concentration of the system are conducive to the oxidation and degradation of TC. The nano-fiber catalytic membrane has been tested for five time cycles and found to have excellent photocatalytic stability and reusability. The study of catalytic mechanism shows that h^+^, •OH and •O_2_^−^ are produced and participated in the oxidation degradation reaction of TC, and •O_2_^−^ plays a major role in catalysis.

## Figures and Tables

**Figure 1 molecules-28-02647-f001:**
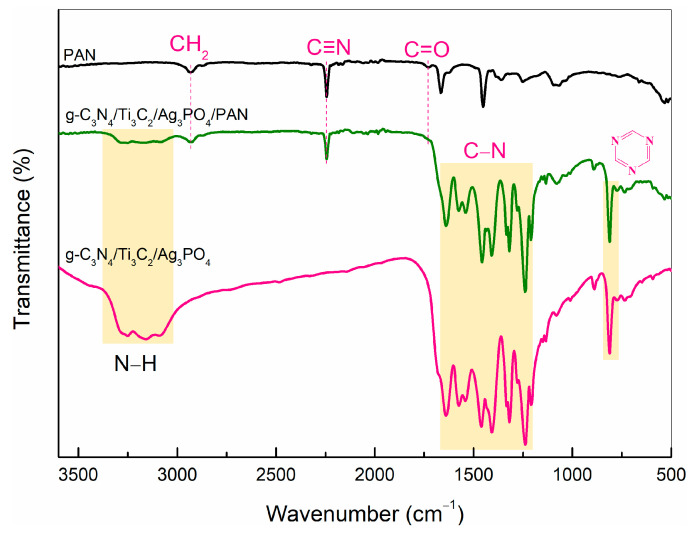
FTIR spectra of PAN, g-C_3_N_4_/Ti_3_C_2_/Ag_3_PO_4_ and g-C_3_N_4_/Ti_3_C_2_/Ag_3_PO_4_/PAN.

**Figure 2 molecules-28-02647-f002:**
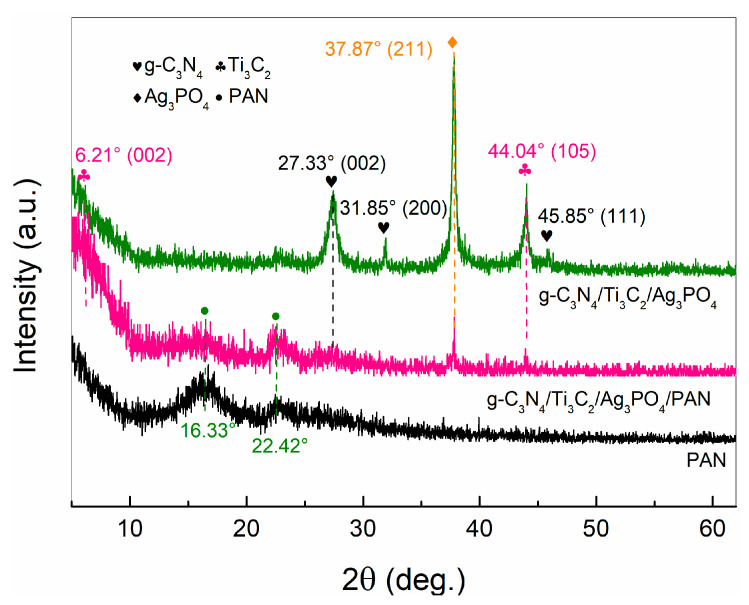
The XRD pattern of PAN, g-C_3_N_4_/Ti_3_C_2_/Ag_3_PO_4_ and g-C_3_N_4_/Ti_3_C_2_/Ag_3_PO_4_/PAN.

**Figure 3 molecules-28-02647-f003:**
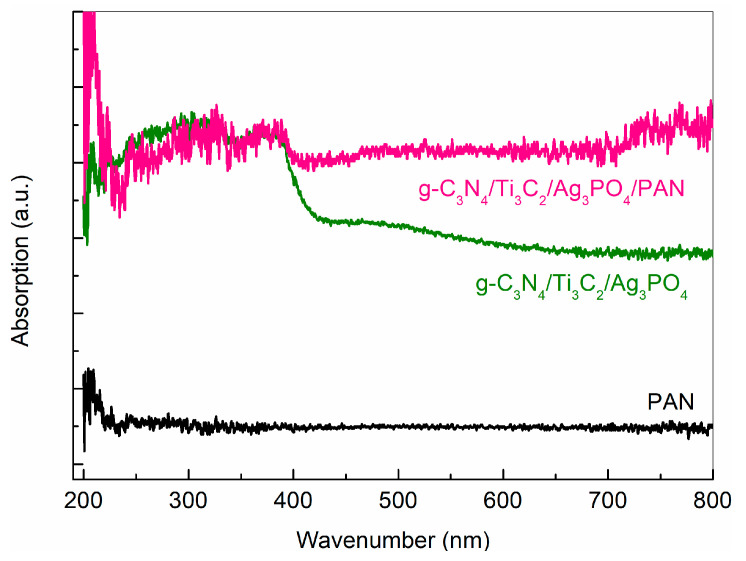
DRS spectrum of PAN, g-C_3_N_4_/Ti_3_C_2_/Ag_3_PO_4_ and g-C_3_N_4_/Ti_3_C_2_/Ag_3_PO_4_/PAN.

**Figure 4 molecules-28-02647-f004:**
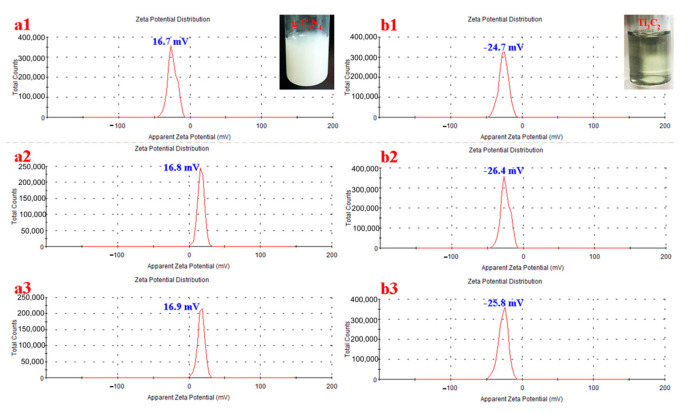
Zeta potential of g-C_3_N_4_ (**a1**–**a3**) and Ti_3_C_2_ (**b1**–**b3**) aqueous dispersion.

**Figure 5 molecules-28-02647-f005:**
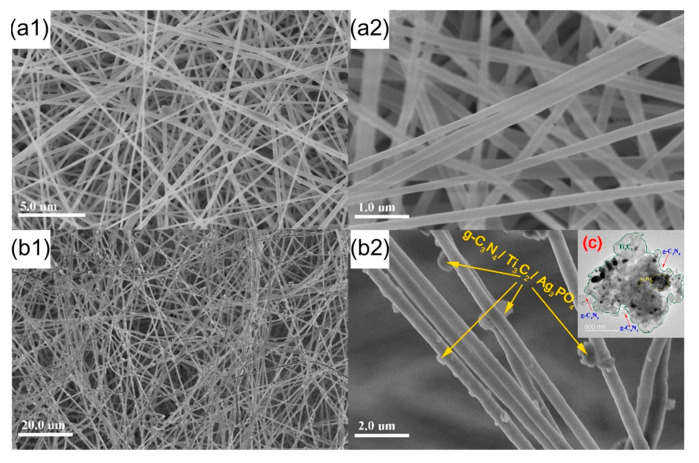
SEM of PAN nanofiber (**a1**,**a2**), PAN nanofiber composite catalytic membrane (**b1**,**b2**), TEM of g-C_3_N_4_/Ti_3_C_2_/Ag_3_PO_4_ catalyst (**c**).

**Figure 6 molecules-28-02647-f006:**
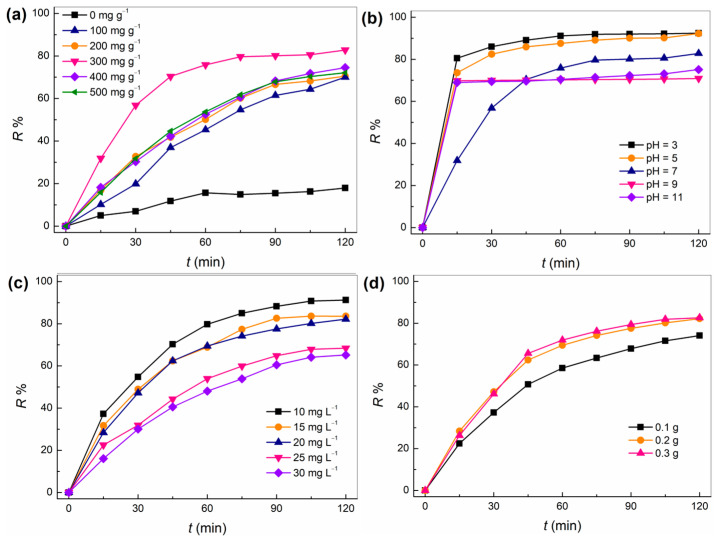
Effect of nanofiber photocatalysis film on catalytic performance: (**a**) loading amount, (**b**) pH, (**c**) TC concentration, and (**d**) fiber film dosage.

**Figure 7 molecules-28-02647-f007:**
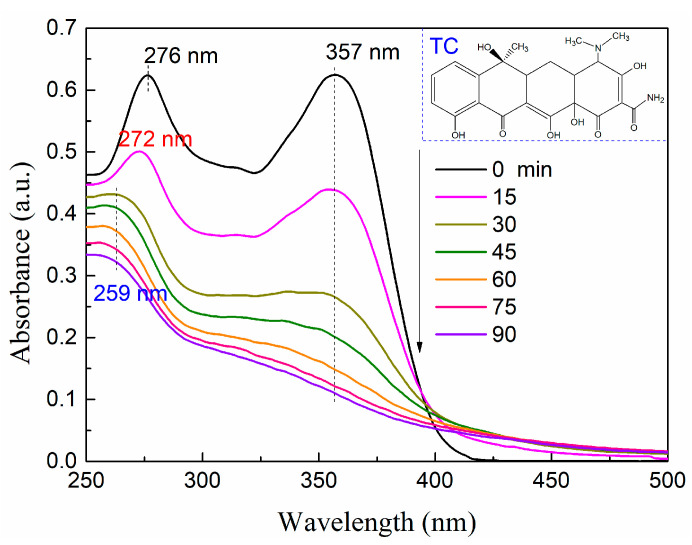
UV spectrum of oxidation degradation of TC in the presence of composite nanofiber membrane.

**Figure 8 molecules-28-02647-f008:**
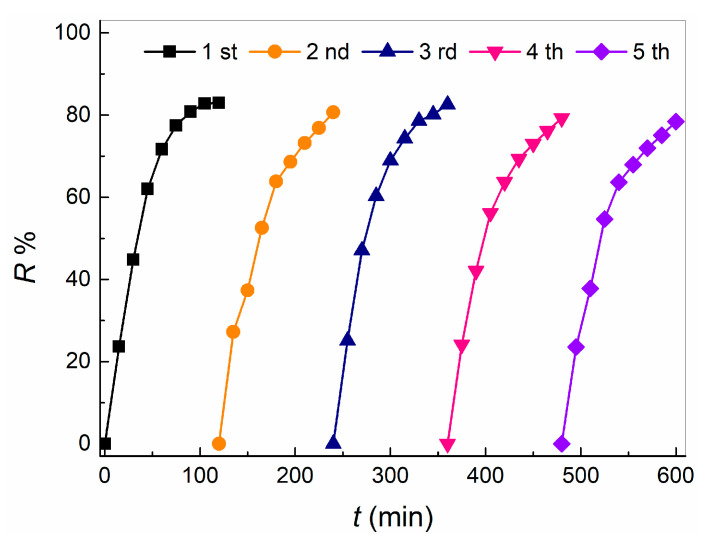
Reuse performance of composite nanofiber membrane in catalytic oxidation degradation of TC.

**Figure 9 molecules-28-02647-f009:**
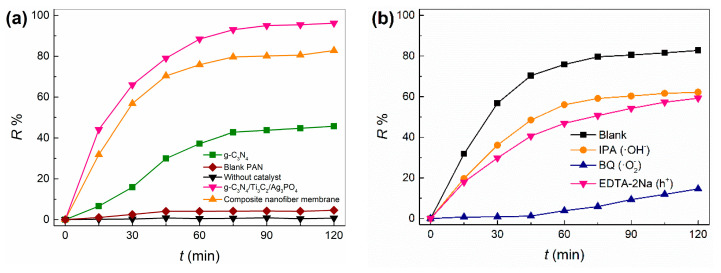
(**a**) Photocatalytic performance of catalyst, (**b**) Effect of catalyst free radical catcher on catalytic performance of nanofiber membrane.

**Figure 10 molecules-28-02647-f010:**
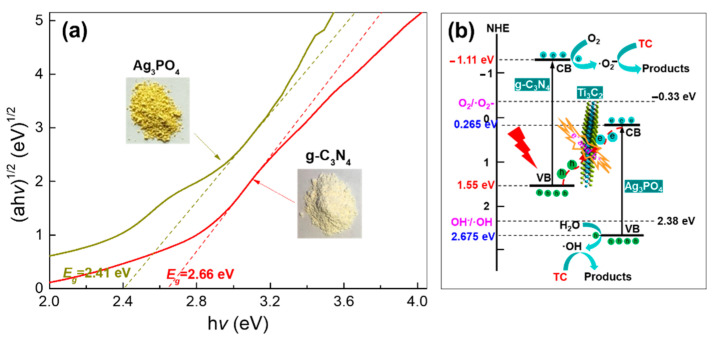
(**a**) The band gap width of g-C_3_N_4_ and Ag_3_PO_4_, (**b**) The mechanism of catalytic oxidation degradation of TC by composite nanofibers.

## Data Availability

Not applicable.

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
