# Peer review of "Removal of Tetracycline Hydrochloride by Photocatalysis Using Electrospun PAN Nanofibrous Membranes Coated with g-C3N4/Ti3C2/Ag3PO4"

_molecules, 2023, doi:10.3390/molecules28062647_

Round 1

Reviewer 1 Report

In this manuscript, the authors investigated the as-prepared g-C3N4/Ti3C2/Ag3PO4 composite heterostructure, and then the g-C3N4/Ti3C2/Ag3PO4/PAN composite nanofiber membrane was prepared by electrospinning technology for photocatalytic degradation of tetracycline hydrochloride (TC). The composite nanofiber membrane has better efficiency and reusability proved by some degradation experiments and a series of characterization techniques. This work shows some new insights into nanofiber-based photocatalysts design. However, what is not satisfactory is that there are still some problems remain to be solved. In my opinion, this article can be accepted for publication after revisions, please refer to the following for specific opinions.

1. The BET of the g-C3N4/Ti3C2/Ag3PO4 and composite nanofiber membrane is not characterized, which has an important impact on photocatalytic activity.

2. In the process of photocatalysis, the g-C3N4/Ti3C2/Ag3PO4 would produce free radicals with strong oxidation performance, which has a strong decomposition ability to TC, and will is it the carrier material PAN nanofiber membrane degraded?

3. How is the loading amount of the nanofiber membrane determined?

4. The adsorption of TC onto the photocatalysts should be measured since it has an important effect on the photodegradation of TC. And the blank experiment without a photocatalyst should be also examined.

Author Response

07-Mar-2023

Dear Editor

Molecules-2283349

Thank you for your letter and for the reviewers’ comments concerning our manuscript ID Molecules-2283349 entitled " Removal of Tetracycline Hydrochloride by Photocatalysis Using Electrospun PAN Nanofibrous Membranes Coated with g-C3N4/Ti3C2/Ag3PO4".

We consider the comments from you and reviewers to be constructive and would like to thank you for your hard effort. In the revised paper, we have done our best to accommodate comments and suggestions from you and reviewers. The corrected sentences have been marked in RED in the revised manuscript. A detailed reply list to comments is shown below.

We hope our revised paper is acceptable to the reviewers and you, and look forward to hearing from you in due course.

Best regards,

Dr. Zhiqi Zhao; Prof. Changlong Li

School of Textile and Garment

Anhui Polytechnic University

Address: No.8, Middle Beijing Road, Jiujiang District, Wuhu

Post code: 241000

E-mail: zhaozhiqi@mail.ahpu.edu.cn; licl@ahpu.edu.cn

Responds to the editor and reviewer’s comments:

Reviewer #1:

In this manuscript, the authors investigated the as-prepared g-C3N4/Ti3C2/Ag3PO4 composite heterostructure, and then the g-C3N4/Ti3C2/Ag3PO4/PAN composite nanofiber membrane was prepared by electrospinning technology for photocatalytic degradation of tetracycline hydrochloride (TC). The composite nanofiber membrane has better efficiency and reusability proved by some degradation experiments and a series of characterization techniques. This work shows some new insights into nanofiber-based photocatalysts design. However, what is not satisfactory is that there are still some problems remain to be solved. In my opinion, this article can be accepted for publication after revisions, please refer to the following for specific opinions.

  1. The BET of the g-C3N4/Ti3C2/Ag3PO4 and composite nanofiber membrane is not characterized, which has an important impact on photocatalytic activity.

A: Specific surface areas (BET) of blank PAN, g-C3N4/Ti3C2/Ag3PO4 and g-C3N4/Ti3C2/Ag3PO4/PAN have been determined by N 2 adsorption using Micromeritics ASAP 2460 equipment. And BET of three samples are10.0041m²/g, 81.3913 m²/g and 20.0023 m²/g respectively. This shows that the BET of composite nanofiber membrane has increased and has an impact on the photocatalytic performance of the composite fiber. And the results had been supplemented in the right place in the paper (line 246).

  1. In the process of photocatalysis, the g-C3N4/Ti3C2/Ag3PO4 would produce free radicals with strong oxidation performance, which has a strong decomposition ability to TC, and will is it the carrier material PAN nanofiber membrane degraded?

A: According to the literature [Yunjin Zhong, Haixiang Chen, Xiufang Chen, et al. Abiotic degradation behavior of polyacrylonitrile-based material filled with a composite of TiO2 and g-C3N4 under solar illumination. Chemosphere 299 (2022) 134375], with the extension of light duration, PAN will be destroyed by the free radicals produced by the catalyst, and the mass loss of PAN is about 60% when the light is 60h. However, for this paper, first of all, the composite nanofiber membrane is cut and used in the process of use. The experimental scale is small, and it is difficult to test its mechanical properties. In addition, the fiber membrane is not stressed during the experimental process. Secondly, the experimental period is short, and the mass loss of PAN can be almost ignored at the end of the test. Furthermore, when the mechanical properties of PAN are greatly lost due to excessive reuse, a new composite nanofiber membrane can be prepared by re-dissolving the composite nanofiber membrane in DMF solution and electrospinning technology.

  1. How is the loading amount of the nanofiber membrane determined?

A: The author controls the content of g-C3N4/Ti3C2/Ag3PO4 in the spinning solution preparation process to control the loading amount of the nanofiber membrane.

  1. The adsorption of TC onto the photocatalysts should be measured since it has an important effect on the photodegradation of TC. And the blank experiment without a photocatalyst should be also examined.

A: The adsorption of TC onto the photocatalysts and blank experiment have been measured, and relevant contents have been added in Figure 9(a).

Reviewer 2 Report

Recommendation: minor revision

Comments:

In the manuscript entitled "Removal of Tetracycline Hydrochloride by Photocatalysis Using Electrospun PAN Nanofibrous Membranes Coated with g-C3N4/Ti3C2/Ag3PO4", Changlong Li, and Zhiqi Zhao et. al., prepared g-C3N4/Ti3C2/Ag3PO4 S-type heterojunction catalyst to improve the photocatalytic performance of g-C3N4. The morphology and chemical properties of the nanofiber membrane were characterized by SEM, FTIR, and XRD, and the photocatalytic degradation of tetracycline hydrochloride (TC) was investigated. The nano-fiber catalytic membrane had been recycled for five times and found to have excellent photo-catalytic stability and reusability. The authors finally proposed that the work provides a new insight into the construction of high-performance and high-stability photocatalytic system by electrospinning technology. The English language needs to be polished before considering the manuscript for publication. Also there are some minor comments that can be addressed while revising the manuscript:

1. Rewrite the 1st sentence of the introduction part and remove the word “abused”

2. Rephrase: “Therefore, it is an urgent problem for scientists to find an efficient method to treat residual antibiotics in surface water, groundwater and even drinking water.”

3. Photo-catalytic can be replaced by ‘Photocatalytic’

4. “At present, the photo-catalysts used mainly include TiO2, Ag3PO4”. There are some other photocatalysts (viz ZnIn2S4) that authors should mentioned in this line. The author should add the reference https://doi.org/10.1016/j.jcis.2022.07.107

5. Before discussing about “g-C3N4 is a new metal-free polymer visible light-driven semiconductor photo-catalyst”, the authors should incorporate one or two sentences regarding the advantages of metal-free organic photocatalysts. Herein, the authors should cite a recently published article “J. Am. Chem. Soc. 2023, 145, 6, 3535–3542” introducing the importance of organic photocatalysts.

6. Results and discussion section should begin with some basic characterization techniques such as FTIR, PXRD, DRS, etc then zeta followed by SEM and so on.

7. Overall the characterization data are good. Though, DRS spectra can be smooth enough to get the sufficient information. The authors can check and cite the reference https://doi.org/10.1016/j.jcat.2020.06.005

8. There is a typo on line 186 page 6.

9. The authors have used BQ are the trapping agents but a reference need to be cited. The authors can follow J. Am. Chem. Soc. 2023, 145, 6, 3535–3542.

Author Response

07-Mar-2023

Dear Editor

Molecules-2283349

Thank you for your letter and for the reviewers’ comments concerning our manuscript ID Molecules-2283349 entitled " Removal of Tetracycline Hydrochloride by Photocatalysis Using Electrospun PAN Nanofibrous Membranes Coated with g-C3N4/Ti3C2/Ag3PO4".

We consider the comments from you and reviewers to be constructive and would like to thank you for your hard effort. In the revised paper, we have done our best to accommodate comments and suggestions from you and reviewers. The corrected sentences have been marked in RED in the revised manuscript. A detailed reply list to comments is shown below.

We hope our revised paper is acceptable to the reviewers and you, and look forward to hearing from you in due course.

Best regards,

Dr. Zhiqi Zhao; Prof. Changlong Li

School of Textile and Garment

Anhui Polytechnic University

Address: No.8, Middle Beijing Road, Jiujiang District, Wuhu

Post code: 241000

E-mail: zhaozhiqi@mail.ahpu.edu.cn; licl@ahpu.edu.cn

Responds to the editor and reviewer’s comments:

Reviewer #2:

In the manuscript entitled "Removal of Tetracycline Hydrochloride by Photocatalysis Using Electrospun PAN Nanofibrous Membranes Coated with g-C3N4/Ti3C2/Ag3PO4", Changlong Li, and Zhiqi Zhao et. al., prepared g-C3N4/Ti3C2/Ag3PO4 S-type heterojunction catalyst to improve the photocatalytic performance of g-C3N4. The morphology and chemical properties of the nanofiber membrane were characterized by SEM, FTIR, and XRD, and the photocatalytic degradation of tetracycline hydrochloride (TC) was investigated. The nano-fiber catalytic membrane had been recycled for five times and found to have excellent photo-catalytic stability and reusability. The authors finally proposed that the work provides a new insight into the construction of high-performance and high-stability photocatalytic system by electrospinning technology. The English language needs to be polished before considering the manuscript for publication. Also there are some minor comments that can be addressed while revising the manuscript:

  1. Rewrite the 1stsentence of the introduction part and remove the word “abused”.

A: The 1st sentence of the introduction part has been rewritten.

  1. Rephrase: “Therefore, it is an urgent problem for scientists to find an efficient method to treat residual antibiotics in surface water, groundwater and even drinking water.”

A: The sentence of “Therefore, it is an urgent problem for scientists to find an efficient method to treat residual antibiotics in surface water, groundwater and even drinking water.” has been rephrased.

  1. Photo-catalytic can be replaced by ‘Photocatalytic’

A: Photo-catalytic has been replaced by ‘Photocatalytic’ in paper.

  1. “At present, the photo-catalysts used mainly include TiO2, Ag3PO4”. There are some other photocatalysts (viz ZnIn2S4) that authors should mentioned in this line. The author should add the reference https://doi.org/10.1016/j.jcis.2022.07.107

A: The photocatalysis-related papers are cited in paper as reference [5].

  1. Before discussing about “g-C3N4 is a new metal-free polymer visible light-driven semiconductor photo-catalyst”, the authors should incorporate one or two sentences regarding the advantages of metal-free organic photocatalysts. Herein, the authors should cite a recently published article “J. Am. Chem. Soc. 2023, 145, 6, 3535–3542” introducing the importance of organic photocatalysts.

A: The article [J. Am. Chem. Soc. 2023, 145, 6, 3535–3542] has been cited in the right place in the paper as reference [6].

  1. Results and discussion section should begin with some basic characterization techniques such as FTIR, PXRD, DRS, etc then zeta followed by SEM and so on.

A: The results and discussion section has been rewritten to make it more logical.

  1. Overall the characterization data are good. Though, DRS spectra can be smooth enough to get the sufficient information. The authors can check and cite the reference https://doi.org/10.1016/j.jcat.2020.06.005

A: Thank you for your review to this paper. The article has been cited in the right place in the paper as reference [23].

  1. There is a typo on line 186 page 6.

A: Apologized for the linguistic errors, after multiple times carefully reading of the whole sentence, all the typos and misspellings were corrected thoroughly.

  1. The authors have used BQ are the trapping agents but a reference need to be cited. The authors can follow J. Am. Chem. Soc. 2023, 145, 6, 3535–3542.

A: The reference has been cited in the right place.